# Let the LLM Stick to Its Strengths:
# Learning to Route Economical LLM

**Yi-Kai Zhang**[1,2]   **Shiyin Lu**[3]   **Qing-Guo Chen**[3]   **Weihua Luo**[3]
**De-Chuan Zhan**[1,2]   **Han-Jia Ye**[1,2*]
[1]School of Artificial Intelligence, Nanjing University
[2]National Key Laboratory for Novel Software Technology, Nanjing University
[3]AI Business, Alibaba Group

## Abstract

Recently, test-time scaling of Large Language Models (LLMs) has emerged as a practical alternative to parameter and data scaling. Reasoning tasks often require large-scale, RLVR-based LLMs, while more economical LLMs can handle simpler tasks. Routing an LLM tailored to *suitability* (*i.e.*, capability and cost) ensures usability and efficiency. We introduce LLMRec, which routes the most suitable LLM to the user query without pre-inference on the candidate LLM zoo. It pioneeringly reframes the LLM routing problem as a comprehensive recommendation system (RecSys) task. Our core insight is that an LLM's suitability for a query is a complex, latent signal equal to user-item preference. LLMRec systematically engineers features for candidate LLMs (intrinsic attributes and capability distributions), queries (general semantics and meta-dimensional info), and context (inference type, cost budgets). It also incorporates behavioral features to learn high-order interactions. LLMRec is designed to generalize to out-of-domain datasets and adapt to new LLMs as the model zoo evolves. We define the metric with the Pareto frontier under user-specified cost budgets. Across six datasets, LLMRec achieves an average cost reduction of over 38% while maintaining accuracy and consistently outperforming baselines in converging toward the Pareto frontier.

## 1   Introduction

The rapid growth of large language models (LLMs) [27, 38] has produced a diverse ecosystem of models varying in scale, functionality, and performance. The parameter scaling era gave rise to models ranging from 1B-parameter edge-device variants to 100B+ omni-models [48, 57]. The subsequent test-time scaling era introduced advanced reasoning LLMs like DeepSeek-R1 [15], which leverage reinforcement learning and extended thinking processes for superior performance.

In practice, companies often provide LLMs-as-API-services [19]. For medium-sized business clients, the monthly expenses for these services can reach millions of dollars [1], [53]. A significant challenge arises from this diversity: the capabilities of large or reasoning-heavy LLMs often far exceed the requirements of some downstream tasks, leading to unnecessary costs. Smaller-scale LLMs are typically sufficient for handling simple tasks, but more complex tasks, such as code generation [10] or mathematical reasoning [12], often require techniques like chain-of-thought (CoT), or larger, reasoning LLMs with extended inference tokens [28, 13, 54]. An intuitive idea emerges: can we intelligently route user queries to the most suitable LLM, balancing performance and cost?

A good model routing outcome hinges on cost considerations, primarily driven by two factors [3]: the cost per token and the number of tokens generated. The former reflects the expense of producing

---

[*]Corresponding author, email: yehj@lamda.nju.edu.cn.

39th Conference on Neural Information Processing Systems (NeurIPS 2025).

a single token, typically scaling with the size of the candidate LLM. The latter is closely tied to the inference mode, or whether the LLM is RLVR-based. The total cost can be defined as the product of these two. A general observation is that larger LLMs or long-chain inference increase costs but improve task completion and accuracy. From that, our model routing goal is to minimize costs without sacrificing accuracy, approaching the Pareto optimum in the trade-off between cost and accuracy.

In this paper, we propose LLMRec, which pioneeringly reframes the model routing problem as a comprehensive recommendation system (RecSys) task. We find the underlying logic of these two problems is equivalent: (**1**) **RecSys** learns the complex match between a user and an item to predict a personalized *Preference*. (**2**) **Model Routing** learns the optimal match between a query and an LLM to evaluate its comprehensive *Suitability* (*i.e.*, capability and cost). Importantly, both tasks share an assumption: RecSys is effective because a user's preference is a complex, latent signal. Similarly, an LLM's suitability for a query is a profound, implicit match determined by its specific capabilities, domain expertise, and cost, which can be learned from massive historical performance logs.

Specifically, we introduce the feature engineering paradigm from RecSys into model routing. In LLMRec, *queries* and *LLMs* correspond to *users* and *items*. We systematically construct features for: (**1**) Model and Query: An LLM representation captures intrinsic attributes (like architecture, scale, origin, training details, *etc.*) and capability distributions (as performance on benchmarks at release and on a core set we constructed). For query representation, we consider general semantic embeddings and meta-dimensional information (including high-level evaluations like answer difficulty, reasoning level, and domain category). (**2**) Context: As RecSys dynamically adjusts recommendations based on different times or locations, we use the user-specified infer type (*e.g.*, self-consistency, CoT, and Tree-of-Thought) and cost threshold as contextual features, enabling adaptation to different service-level objectives. (**3**) Behavioral sequences: Furthermore, we build dynamic interaction features based on the performance of candidate LLMs on **i**) a core set of tasks and **ii**) on top-k nearest training neighbors of the current query. Like modern RecSys, the core of the LLMRec framework learns the complex, high-order cross-information among model, query, behavioral, and contextual features. This drives it to accurately predict a *suitability* score for each candidate LLM, recommending the one that best approaches the Pareto optimum for the given cost constraint.

LLMRec is designed to be robust, generalizing to out-of-domain datasets, and adapting to new LLMs as the model zoo evolves. While maintaining accuracy, LLMRec reduces costs by an average of over 38% across six datasets. It consistently outperforms all baselines in converging toward the Pareto frontier under various cost budgets. Our main contributions are as follows:

- We are the first to systematically apply recommendation systems to model routing, achieving economical, efficient, and iterative optimization.
- We implemented model, query, behavioral, and contextual features, allowing users to specify the inference type and a cost threshold to constrain the routing.
- We develop a cost-budget-based Pareto metric tailored for LLM routing applications.
- We construct a large-scale training set and routing benchmark, including evaluations on unseen datasets and dynamic model zoos.

LLMRec is versatile, efficient, and adaptable to the evolving LLM landscape, offering a scalable solution for real-world LLM API deployment.

## 2 Preliminary

We start by presenting the model, query, and pipeline of the LLM router, and discuss its connection to RecSys. We then introduce the deployment and related work of model routing.

### 2.1 Notations

**Key Elements in LLM Routing** Consider a scenario where, when using an LLM API, the system provides a candidate LLM zoo $\mathcal{M} = (f^1, f^2, ..., f^M)$. In this case, the user provides a task consisting of an instruction set $\mathcal{D}_{\text{test}} = \{(\mathbf{x}_i, \mathbf{a}_i)\}_{i=1}^N$, where the LLM $f^m$ produces the output as $\mathbf{o}_i = f^m(\mathbf{x}_i; \mathbf{I}_j)$ on input $\mathbf{x}_i$ using inference mode $\mathbf{I}_j \in \mathcal{I}$, and the correct answer is $\mathbf{a}_i$. The accuracy is given by $\text{Acc}(\mathbf{o}_i, \mathbf{a}_i)$. In this paper, we focus on the case that $f^m$ represents a decoder-only text generation LLM. As established in the section 1, the core challenge of LLM routing is to predict the *suitability* of an LLM $f^m \in \mathcal{M}$ for a given query $(\mathbf{x}_i; \mathbf{I}_j)$ *without* first generating the actual output $\mathbf{o}_i$

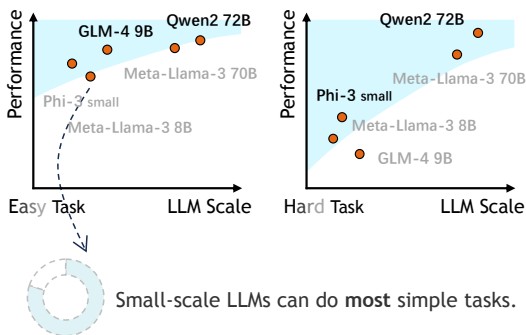

Figure 1: **Comparison of the performance laws** of LLMs with different scales, architectures, and tasks. For most LLM architectures, smaller-scale LLMs can accomplish most simple tasks efficiently; routing to a smaller-scale LLM can reduce deployment costs.

from each candidate model. This *suitability* is conceptually consistent with the foundation of RecSys, which operates by predicting the complex user *preference*.

**Brief Background on Recommendation System**    The core task of RecSys is to learn the complex matching relationships between *users* and *items*. A typical process begins by collecting user and item features, their historical interaction logs, and contextual information. The model then learns a user's preference for different items under various scenarios. The ranking task in RecSys estimates the probability of a user clicking on each item.

**Cost-effective Routing Pipeline**    Our task is to select an LLM $f^m$ for each instruction $\mathbf{x}_i$ and the inference mode $\mathbf{I}_j$ to minimize cost while maintaining accuracy. We define the per-token cost of an LLM as $t^m$, and the goal is:

$$\boldsymbol{f}^* = \underset{f^m \in \mathcal{M}, \mathbf{I}_j \in \mathcal{I}}{\arg\max} \ \mathbb{E}_i \left[ \mathrm{Acc} \left( f^m \left( \mathbf{x}_i; \mathbf{I}_j \right), \mathbf{a}_i \right) \right], \quad \text{s.t.} \ \sum_{i=1}^n t^m \cdot |\mathbf{o}_i| \leqslant \epsilon \,, \tag{1}$$

where $\mathbf{o}_i = f^m \left( \mathbf{x}_i; \mathbf{I}_j \right)$. The cost of the input sequence is omitted (as it is fixed for all methods). $\epsilon$ represents the user-specified cost threshold. In our formulation, exceeding the cost threshold $\epsilon$ incurs some penalty, but it is less critical than losing accuracy. The upper bound of LLM routing performance depends on the number of instructions for which no available LLM can produce a correct answer. In practice, directly optimizing Equation 1 as a hard-constraint problem is intractable. The budget $\epsilon$ is a *global* constraint summed over the entire dataset, whereas the router must make a *local* decision for each query $\mathbf{x}_i$. Furthermore, a router trained to satisfy a fixed $\epsilon$ is inflexible; it cannot adapt to a different user-specified budget without being retrained. To overcome this, we reframe the problem by treating the cost threshold $\epsilon$ as a dynamic input feature. This converts the hard constraint into a learnable condition. The router learns a policy that associates different budget levels with corresponding routing strategies, enabling it to dynamically balance cost and accuracy based on the user's needs. We formally define the router, $f^{\mathrm{router}}$, as a policy that selects a model $f^m \in \mathcal{M}$ given an input $\mathbf{x}_i$, an inference mode $\mathbf{I}_j$, and the cost threshold $\epsilon$.

**Definition of the Pareto Front based on Cost Intervals**    In practical applications, service providers tend to prioritize accuracy as a key performance indicator over cost. This preference stems from their desire for a predictable performance outcome rather than an uncertain level of accuracy after incurring expenses. Therefore, when LLM API service providers implement LLM routing, they often address the dual problem of the optimization presented in the above Equation. This dual problem aims to minimize costs subject to a constraint on a target accuracy. At this point, we introduce the concept of Pareto dominance. For any two routing solutions, say solution $f_a^{\mathrm{router}}$ and solution $f_b^{\mathrm{router}}$, characterized by their respective metrics (*e.g.*, accuracy and cost), we state that solution $a$ Pareto dominates solution $b$ (denoted as $f_a^{\mathrm{router}} \succ f_b^{\mathrm{router}}$) if and only if solution $a$ is strictly better than solution $b$ in at least one objective and not worse in the other objectives. If one solution has lower accuracy but also a lower cost than another, neither solution dominates. They may belong to the same

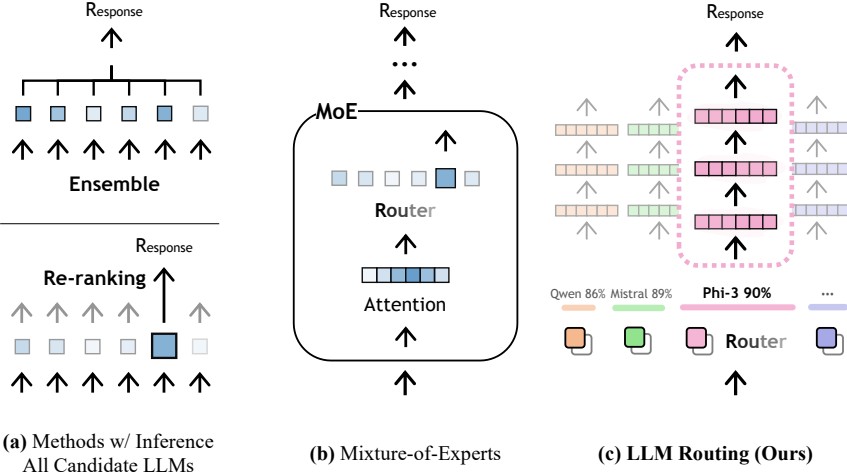

**(a)** Methods w/ Inference All Candidate LLMs

**(b)** Mixture-of-Experts

**(c)** LLM Routing (Ours)

Figure 2: **Comparison of Different Methods for Utilizing pre-trained LLM Libraries**: Ensemble and Re-ranking methods require inference from all candidate LLMs. The Ensemble approach combines the outputs of multiple LLMs, while the Re-ranking selects the most suitable response from all generated outputs. Mixture-of-Experts (MoE) integrates routing within the Transformer layer structure. LLM Routing in our setup directs instruction to one of the candidate LLMs without requiring inference on the target instruction for all LLMs.

Pareto front. If both solutions fall within the user's acceptable range for both cost and accuracy, they are both considered comparatively optimal.

## 2.2 Related Work

Leveraging the formalized setting, we revisit key features from the section 1:

- **No pre-inference model interactions.** Some model ensemble or cascade strategies aim to se-lect and synthesize the optimal response from all candidate models, considering input-response relationships. Some scoring strategies have been applied in reinforcement learning-based LLM training. Others include classical machine learning techniques [7, 40, 20, 21] and multi-perspective deep ensemble methods [35, 22]. However, these methods can introduce significant delays in LLM APIs, especially with large-scale candidates. In LLMRec, inferring each model for every target instruction is not feasible.
- **Generalization in new scenarios.** LLMRec is designed for LLM APIs with a focus on router scalability. While some transfer learning approaches [5, 29, 59, 39] use proxy source-target metrics based on label distribution matching, they are limited by the target set seen during training [9]. On the other hand, LLM APIs have a flexible natural language output space, meaning the router should generalize zero-shot to unseen user instructions.
- **Extension on updated candidate models.** Some application-specific router frameworks [44, 3, 14] have addressed the issues mentioned above. However, most approaches fix the candidate model zoo in order to stabilize routing training and deployment scenario [36, 16, 17, 46, 37]. Additionally, some mixture-of-experts (MoE) [45, 18, 31] use MLPs within transformer blocks as experts, embedding a router to select among them to reduce inference costs. However, these methods tightly couple router and model parameters, so when the expert zoo is updated, the router needs to adapt through complex incremental learning [41, 64], which may introduce issues like hyperparameter sensitivity and catastrophic forgetting.

In summary, while existing approaches have made significant strides in specific aspects of model routing, LLMRec distinguishes itself by offering a solution that holistically addresses the challenges of pre-inference overhead, zero-shot generalization to new instructions, seamless extension to updated model libraries, and inherent cost-efficiency. Its novel use of a recommendation system framework, as depicted in Figure 3, based on learnable representations of models and queries, allows for dynamic and efficient routing in large-scale LLM API environments. LLMRec introduces a scalable approach

by creating a universal, learnable model representation, turning LLM routing into learning both model and query embeddings. The model representation encodes capabilities and behaviors and is optimized with a dynamic embedding vocabulary. This universality allows new LLMs to quickly index into the embeddings after lightweight inference, unlike the random initialization and re-training needed in Model Spider [63]. Finally, the routing process estimates the relationship between the model and query representations.

## 3 Learning to Route Test-Time Economical LLM

In this section, we outline LLMRec framework, discuss the construction of the recommendation representations, implementation details, and cost constraints in deployment.

### 3.1 Representation of Model & Query

**Motivation**: The model is a black box to the router, and decisions must be made without inferring on all candidate models to minimize overhead. Directly extracting features from an LLM's high-dimensional parameters is infeasible. To address this, we construct a *model representation* that incorporates both *intrinsic properties* and *capability distributions*. This allows the router to learn how a model's potential impacts its generalization to new instructions. To achieve this, we also build a comprehensive *coreset* of diverse evaluation data. The advantage is that when a new model arrives, we can assess its core capabilities on this coreset with minimal overhead.

**Model Representation**   For the candidate LLM $f^m$, we categorize its representation into intrinsic properties and capability distributions, with each dimension optional.

1. **Intrinsic properties** include model structure (*e.g.*, *publisher*, *name*, *architecture*, *number of layers*, *layer types*, *total parameters*, *training details*, *precision*, and *feature descriptions*). Additional information may include *HuggingFace download count*, *open-source licenses*, etc.
2. **Capability distributions** are divided into evaluated on offline benchmark and online coreset.
   (a) Given that most LLMs publish standard benchmark performances on release, we document model performance on benchmarks such as MMLU [24], MMLU-Pro [51], BBH [47], ARC-Challenge [6], TruthfulQA [32], Winogrande [43], and HellaSwag [60]. For reasoning capabilities, we consider domains like mathematics (MATH [25], MMLU-STEM, GSM8K [12]) and code generation (HumanEval [10], HumanEval+ [34], MBPP [4], MBPP+ [34]). For offline capabilities, we focus on the average performance across these datasets, and definitely, the router will not explicitly know which benchmark the current user query belongs to.
   (b) To expand the assessment of model capabilities, we also create an online evaluation *coreset*: it acts as a bridge linking the model's historical behavior with its future expected performance. By sampling 20-shot examples from each of the 71 categories in MMLU and MMLU-Pro, we form a core set of 1,415 instructions for online evaluation. In parallel, we select from specialized domains like mathematics, code generation, healthcare, law, and finance. We extract 5 keywords for each category as semantic descriptors and compute class-center embeddings. Detailed information is provided in the section 4.

**Query Representation**   We divide the representation for a user instruction $\mathbf{x}_i$ into general semantic representation and meta-dimensional information.

1. **General semantic embeddings**: For a user instruction $\mathbf{x}_i$, we use 3 general encoders $\psi$ (*e.g.*, GTE$_{large}$ [62] $\sim$ 0.33B, Qwen2.5-0.5B-Instruct [57], and RoBERTa-Large) to extract embeddings.
2. **Meta-dimensional information**, *e.g. answer difficulty*, *reasoning level*, *content diversity*, *temporal stability*, *conceptual ambiguity*, and *domain expertise*, is extracted via specific prompts by encoder Qwen2.5-7B-Instruct [57] through a few inference steps.

**Context Representation**   We also construct contextual features to align routing decisions with service-level objectives. These include the user-specified *inference type* (*e.g.*, self-consistency, CoT, Tree-of-Thought (ToT)) and the *cost threshold*, which we discretize into five levels.

The features described above are all attributes of either the query or the model side. In RecSys, these are known as first-order features. We note that any of these feature dimensions can be optional (in implementation, they are null-padded). Modern RecSys, however, automatically constructs and learns

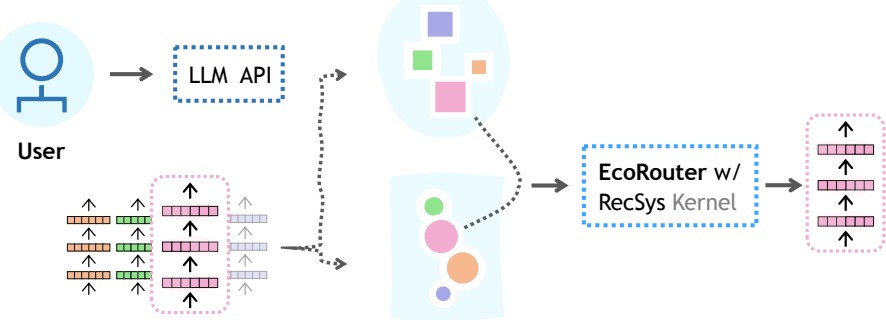

Figure 3: **The flowchart of LLMRec.** The representation of model, query, context, and behavioral sequences is constructed. Then, the feature relationships are learned by the LLMRec with the RecSys kernel to route to the corresponding LLM.

from numerous high-order cross-features to capture the complex, non-linear matching relationship between the two sides. We introduce some examples of *explicit* cross-features, such as those centered on the historical behavioral sequences of models relevant to the current query. Specifically, we consider the performance of candidate LLMs on **i**) our core set of tasks and **ii**) on the top-k nearest training neighbors of the current query. These are introduced as dynamic capability features that are more representative of and relevant to the query. While an inherent gap may exist between a model's static capabilities and a query's semantics, the LLMRec framework, like modern RecSys, is designed to learn from the complex, high-order relationships among all available model, query, contextual, and behavioral sequence features.

## 3.2 Routing as Recommendation

We formalize the routing problem as a ranking in RecSys. In this paradigm, the input query ($\mathbf{x}_i$) and candidate LLM ($f^m$) are mapped to the *user* and *item*. The user-specified context, such as the cost threshold and inference type, provides the situational context for the recommendation. The core objective is to learn a function that predicts the suitability for any given (query, model, context) tuple. This suitability is a complex, non-linear match using high-order cross-features, corresponding to the core challenge in modern RecSys. LLMRec is designed to leverage principles from established architectures to learn these interactions.

For instance, methods like Wide & Deep [11] excel by combining two components: a "wide" part that memorizes explicit, low-order feature interactions (*e.g.*, manually crafted crosses) and a "deep" part that uses MLPs to generalize and learn implicit, high-order relationships from dense embeddings. Manually designing cross-features is difficult. Models like **DeepFM** [23] automate this by integrating Factorization Machines (FM) with an MLP. The FM component efficiently models second-order feature interactions by learning a low-dimensional latent vector for every feature and taking their dot products. These latent vectors are shared with the deep component, allowing end-to-end learning. Enhancements like **AFM** [55] introduce an attention mechanism to weigh the importance of different feature interactions. Other models learn implicit interactions more directly: **FGCNN** [33] applies convolutions to the feature map to capture local interaction patterns. **FiGNN** [30] employs graph neural networks (GNNs), treating features as nodes to capture complex, high-order relationships via aggregation.

Furthermore, a model's suitability is not static; it is highly dependent on the query. We draw inspiration from behavioral modeling in RecSys, such as the Deep Interest Network (DIN) [65], which uses an attention mechanism to dynamically weigh a user's historical behaviors based on their relevance to a target item. Analogously, LLMRec treats a model's historical performance, such as on our *coreset* or on the *top-k nearest training neighbors* of the current query, as a dynamic behavioral sequence. This allows the framework to learn a query-aware representation of a model's capabilities.

By integrating first-order features (of query, model, context) and learning their complex, high-order interactions, the RecSys framework learns to accurately rank all candidate LLMs. This ranking

allows it to recommend the model that best approaches the Pareto optimum for the given query and service-level constraints.

### 3.3 Training Data Construction

A training instance is $(\mathbf{q}, \mathbf{m}, \mathbf{c}, \mathbf{b}) \to s$, where the representations for the query $(\mathbf{q})$, model $(\mathbf{m})$, context $(\mathbf{c})$, and behavioral features $(\mathbf{b})$ map to a target suitability score $s$. To learn this function, we construct a massive dataset of over 1 billion model-query pairs. This dataset spans more than 50 LLM families (from 2024) evaluated on over 30 diverse benchmarks. We capture performance logs using various inference modes, including *direct* generation, *self-consistency* [50], *Chain-of-Thought (CoT)* [52], and *Tree-of-Thought (ToT)* [58].

For each pair $(\mathbf{x}_i, \mathbf{o}_i^m)$ (instruction $\mathbf{x}_i$, model $m$'s output $\mathbf{o}_i^m$), we record the key labels as:

1. Accuracy (Acc): Evaluated using dataset-specific metrics against the ground-truth answer $\mathbf{a}_i$.
2. Cost: Calculated as $\text{Cost}(\mathbf{o}_i^m) = c_{f_m} \times |\mathbf{o}_i^m|$, where $c_{f_m}$ is the estimated per-token inference cost (based on model scale) and $|\mathbf{o}_i^m|$ is the length of response tokens.

To generate the ground-truth suitability ranking $s$, we follow a Pareto-based principle that prioritizes cost-effectiveness. For a given query, we define the preference order as:

$$\text{Ranking} = \underset{\text{Cost}(\mathbf{o}_i^m)}{\text{Sort}} \left( \{f_m\}_{\text{Acc}(\mathbf{o}_i^m, \mathbf{a}_i) > 0} \right) \oplus \text{Shuffle} \left( \{f_m\}_{\text{Acc}(\mathbf{o}_i^m, \mathbf{a}_i) = 0} \right) \tag{2}$$

where $\oplus$ denotes ordered concatenation. All models that correctly solve the task (Acc > 0) are ranked first, sorted by their $\text{Cost}(\cdot)$ in ascending order (lower cost is better). All models that fail (Acc = 0) are shuffled and ranked last. Crucially, this ground-truth ranking is computed relative to the input contextual features $\mathbf{c}$, especially the user-specified *cost threshold*. This teaches the LLMRec to adapt its routings, learning which LLM is "best" under different budget constraints.

## 4 Experiments

### 4.1 Implementation Details

**Candidate LLMs Zoo of Training**    As mentioned, we consider 52 different LLMs. Among them, 32 models are under 10B parameters, 15 models are between 10B and 20B parameters, and 5 models are around 70B parameters. In practice, we test more than 80 models, but exclude early models that did not have CoT capabilities.

**Domains of Training Datasets**    We consider general evaluation datasets, commonsense reasoning, math reasoning, code generation, symbolic reasoning, and specific domain datasets such as medical, law, and financial datasets, totaling 35 datasets.

**Interaction Scaling**    We perform inference with each candidate LLM on the target datasets to generate nearly 10m interaction pairs. We sample approximately 1100k of these for training, with stronger-performing pairs being assigned higher sampling weights. Out of the 35 datasets, 24 are multiple-choice datasets, and 11 are fill-in-the-blank or question-answer datasets, including token generation via the generate method. For these datasets, we incorporate self-consistency, CoT, and Tree-of-Thought (ToT) reasoning modes. We also include datasets where the performance did not improve or even declined after applying complex reasoning modes.

**Evaluation Metrics**    We evaluate the 24 multiple-choice datasets using perplexity (PPL). Most additional 11 fill-in-the-blank datasets are evaluated using regular expressions to extract the final answers. We follow the corresponding evaluation libraries for some domain-specific datasets (such as math extraction processes or code generation-type pass@k).

**Candidate LLMs Zoo of Evaluation**    As shown in Table 1, we have mixed 5 small-scale LLMs with fewer than 10B parameters, 2 LLMs between 10B and 20B parameters, and 3 large-scale LLMs with around 70B parameters to ensure that the LLM library contains varying capabilities, from small to large.

| Method | #Params | General | Comm. Reasoning | | | Mean |
|---|---|---|---|---|---|---|
| | | MMLU 5-shot | TruthfulQA 0-shot | ARC-C 25-shot | MMLU-stem 5-shot | |
| | | | *Small-scale LLMs (<10B)* | | | |
| InternLM2.5 [8] | 7.7B | 69.88 | 54.56 | 60.75 | 65.31 | 62.63 |
| Meta-Llama-3 Instruct [49] | 8.0B | 65.59 | 51.63 | 62.12 | 58.32 | 59.42 |
| Qwen2 Instruct [56] | 7.6B | 69.13 | 55.49 | 61.43 | 63.45 | 62.38 |
| GLM-4 [61] | 9.4B | 69.28 | 59.32 | 66.13 | 64.45 | 64.80 |
| Phi-3 Small-128K [2] | 7.4B | 75.90 | 64.62 | 71.08 | 69.09 | 70.17 |
| Best-Performing of *Small-scale LLMs* | - | 75.90 | 64.62 | 71.08 | 69.09 | 70.17 |
| | | | *Large-scale LLMs (∼70B)* | | | |
| Meta-Llama-3 Instruct [49] | 70B | 79.89 | 61.83 | 71.67 | 73.92 | 71.83 |
| Qwen2 Instruct [56] | 72B | 83.79 | 54.85 | 68.62 | **79.85** | 71.78 |
| Mixtral-8x22B Instruct-v0.1 [26] | 140B | 77.63 | **68.19** | 72.78 | 71.64 | 72.56 |
| Best-Performing of *Large-scale LLMs* | - | 83.79 | 68.19 | 72.78 | 79.85 | 76.15 |
| | | | **LLM Routing** | | | |
| Random Selection | ∼ 32B | 72.98 | 58.87 | 67.83 | 67.96 | 66.91 |
| GTE Large [62] | ∼ 55B | 74.72 | 61.08 | 69.62 | 69.12 | 68.64 |
| **Ours** w/ LR [42] | ∼ 33B | 73.25 | 59.61 | 66.04 | 68.12 | 66.76 |
| **Ours** w/ Deep & Wide [11] | ∼ 27B | 82.32 | 67.44 | 71.67 | 77.37 | 74.70 |
| **Ours** w/ DeepFM [23] | ∼ 26B | 82.29 | 67.81 | 71.93 | 76.74 | 74.69 |
| **Ours** w/ AFM [55] | ∼ **25B** | 79.84 | 63.53 | 70.39 | 77.50 | 72.82 |
| **Ours** w/ DIN [65] | ∼ 31B | **83.92** | 66.83 | **72.95** | 78.36 | **75.52** |

Table 1: **Comprehensive Routing Evaluation on General, Commonsense, and Reasoning Tasks.** We compare the response accuracy across various benchmarks (MMLU, TruthfulQA, ARC-C, and MMLU-stem). We categorize methods by model scale (Small-scale LLMs, Large-scale LLMs, and LLM Routing methods). We show the number of parameters ("#Params"). Bold represents the best performance, and underlined is the second-best.

**Evaluation Benchmarks**  Similarly, we have construct evaluation dataset that includes general evaluation benchmarks like MMLU [24], TruthfulQA [32], and commonsense reasoning tasks such as ARC-Challenge [6] and MMLU-stem [24], as well as mathematical reasoning benchmarks like GSM8K [12], and symbolic reasoning tasks like BBH [47]. Different models exhibit performance variations on these datasets. We reference the scale size and FLOPs to correspond with the cost, and by multiplying the response token count, we calculate the total cost per unit on each dataset.

**Model Representation Construction.** As outlined in section 3, we construct model representations using the descriptions and coreset. Each dimension is segmented into multiple values, with continuous values being bucketed. These segmented values are then mapped to randomly initialized embeddings, which are incorporated into the training process.

**Baseline methods.** For the baseline, we consider randomly selecting from the LLM library and using all responses $\mathbf{o}_i$ from the inference downstream datasets. We employ GTE Large to match $\mathbf{o}_i$ with the instruction $\mathbf{x}_i$, and the LLM with the highest score is selected.

**Average scale calculation** ("#Params" in the Tables). We average the scales of all selected LLMs based on the instruction dimensions of all evaluation data in both tables, which results in the LLM scale presented in "#Params".

**Generalization to unseen models and datasets.** As shown in Table 1, the models underlined in our LLM library have not appeared in the routing training set (one large-scale and one small-scale LLM). All datasets, except for MMLU, are considered unseen datasets.

| Method | #Params | Math Reasoning GSM8K 4-shot, CoT | | Symbolic Reasoning BBH 3-shot, CoT | | Mean | |
|---|---|---|---|---|---|---|---|
| | | Perf. | Leng. | Perf. | Leng. | Perf. | Leng. |
| *Small-scale LLMs (<10B)* | | | | | | | |
| InternLM2.5 | 7.7B | 74.37 | 371 | 68.13 | 452 | 65.50 | 412 |
| Meta-Llama-3 Instruct | 8.0B | 56.18 | 273 | 60.93 | 399 | 59.13 | 336 |
| Qwen2 Instruct | 7.6B | 78.92 | 368 | 62.92 | 526 | 65.22 | 447 |
| GLM-4 | 9.4B | 79.53 | 505 | 74.43 | 487 | 68.86 | 496 |
| Phi-3 Small-128K | 7.4B | 82.34 | 449 | 73.94 | 521 | 72.83 | 485 |
| Best-Performing | - | 82.34 | 449 | 74.43 | 487 | 72.91 | 468 |
| *Large-scale LLMs (∼70B)* | | | | | | | |
| Meta-Llama-3 Instruct | 70B | 83.17 | 580 | 81.48 | 635 | 75.33 | 608 |
| Qwen2 Instruct | 72B | **88.86** | 535 | **82.89** | 593 | 76.48 | 564 |
| Mixtral-8x22B Instruct-v0.1 | 140B | 84.31 | 553 | 79.54 | 610 | 75.68 | 582 |
| Best-Performing | - | 88.86 | 535 | 82.89 | 593 | 79.39 | 564 |
| **LLM Routing** | | | | | | | |
| Random Selection | ∼ 32B | 77.71 | 489 | 73.58 | 534 | 69.82 | 512 |
| GTE Large | ∼ 55B | 80.52 | 528 | 74.37 | 580 | 71.57 | 554 |
| **Ours** w/ LR | ∼ 33B | 83.40 | 457 | 73.84 | 532 | 70.71 | 495 |
| Deep & Wide | ∼ 27B | 86.05 | 426 | 78.27 | 489 | 77.19 | 458 |
| DeepFM | ∼ 26B | 87.87 | 411 | 78.02 | 495 | 77.44 | 453 |
| AFM | **∼ 25B** | 86.05 | 404 | 79.40 | 476 | 76.12 | 440 |
| DIN | ∼ 31B | 87.19 | 446 | 79.94 | 500 | **78.20** | 473 |

Table 2: **Comprehensive Routing Evaluation on Math and Symbolic Reasoning Tasks.** We compare performance on math and symbolic reasoning tasks, showing response accuracy ("Perf.") and average token usage ("Leng."). Similar to Table 1, we estimate the approximate model scale associated with each routing method. The total computational cost of a method on the data is approximately proportional to the product of the model scale ("#Params") and the average token usage ("Leng."). Bold is the best, and underlined is the second-best.

## 4.2 Results Analysis

Table 1 illustrates the performance of LLMRec using different recommendation system kernels on General and Commonsense Reasoning tasks. We test our routers at two scales, with small-scale LLMs showing some variation in performance. Among the small-scale LLMs, Phi-3 Small-128K achieved the best results, scoring the highest across all tests with an average of 70.17. Other small-scale LLMs, such as InternLM2.5, Meta-Llama-3 Instruct, Qwen2 Instruct, and GLM-4, also deliver decent performance.

In the large-scale LLM category, Qwen2 Instruct (72B) outperforms the others, notably earning a high score of 79.85 on MMLU-stem. Meta-Llama-3 Instruct (70B) follows closely, showing strong performance across most tests. Mixtral-8x22B Instruct-v0.1 excels in specific metrics like TruthfulQA, but its overall average score was slightly lower than the other methods. LLMRec manages the routing of instructions for all the above LLMs. Comparison methods were also used, such as random selection and embedding matching using GTE Large (with estimated LLM scales of 32B and 55B). Most of LLMRec's routes were in the 20B+ range. When routing LLMs with the recommendation system's DIN architecture, the best results are achieved when historical model-data interactions are included in the sequence, reaching an average accuracy of 75.52. Wide & Deep and DeepFM also achieve similarly high performance with relatively low overhead. Notably, on MMLU and ARC-Challenge, the LLM's performance surpasses the top-performing models in the datasets.

In Table 2, we present the routing capability on deep reasoning tasks, including GSM8K and BBH datasets for math and symbolic reasoning. The candidate LLM library and comparison methods are the same as in Table 1. In this evaluation, the LLMs performed inference through a generation-based

approach and assembled CoT. The "Mean" in the Table 2 represents the average across all data, including the results from Table 1. After inference, we calculate the average token length for each method. The overall cost for each method is roughly determined by multiplying the model scale by the token length. LLMRec achieves superior performance while reducing the average output length by approximately 30% compared to others, resulting in the optimal average performance.

## Limitations

LLMRec has not considered market factors such as fluctuations in LLM API pricing; instead, we use the LLM parameter scale as an approximation. The generalization capability of model representations under modified pricing schemes requires further investigation. The system's evaluation of new LLMs depends on a sufficient volume of historical interaction data. This creates a cold-start challenge, a common issue for all routing systems in the rapidly evolving LLM ecosystem. This limitation may be addressed by continuously updating the coreset to enhance the validity of the model representations.

## 5 Conclusion

In this paper, we introduced LLMRec, a novel framework that optimizes the cost-performance trade-off for LLM API services by pioneeringly reframing model routing as a recommendation system (RecSys) task. Our core insight is that an LLM's suitability for a query, balancing capability and cost, is a complex, latent signal equal to user-item preference. LLMRec learns this signal by modeling rich, multidimensional features for models, queries, operational contexts (including user-specified cost thresholds), and behavioral sequences. Our extensive experiments demonstrate that LLMRec reduces inference costs by over 38% on average while preserving task accuracy. The framework shows robust zero-shot generalization to unseen tasks and seamless adaptation to a dynamically evolving model zoo, validating intelligent routing as a viable solution for scalable and economical LLM deployment.

## Acknowledgments and Disclosure of Funding

National Key R&D Program of China (2024YFE0202800), NSFC(62376118), Collaborative Innovation Center of Novel Software Technology and Industrialization.

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
