# OpenReview forum: "Let the LLM Stick to Its Strengths: Learning to Route Economical LLM"
_NeurIPS.cc/2025/Conference — NeurIPS 2025 poster_

### Official Review · Reviewer_MPGf · 2025-06-24

**Clarity:** 3
**Significance:** 2
**Originality:** 3
**Rating:** 3
**Confidence:** 3

**Summary:**

This paper introduces EcoRouter, a novel framework for routing user instructions to the most cost-effective Large Language Model (LLM) from a given library, while maintaining high performance. The core problem addressed is the inefficient and expensive use of large, powerful LLMs for tasks that could be handled by smaller, cheaper models. EcoRouter frames this challenge as a recommendation system problem, where the goal is to "recommend" the best LLM (item) for a given instruction (user).

**Questions:**

1. Throughput analysis with other LLMs.
2. Comparison with sota LLMs, e.g., Qwen2.5-14B-1M achieves 94.8 on GSM8K.
3. Does the proposed method achieve steady improvement with more few-shot samples?

**Ethical Concerns:**

["NO or VERY MINOR ethics concerns only"]

**Final Justification:**

The author's rebuttal partly resolved my doubts (I don't see any explanation about Q1 and Q3), and I decided to maintain the previous rating unchanged.

**Limitations:**

Please refer to the questions section.

**Quality:**

3

**Strengths And Weaknesses:**

Strengths

1. The conceptualization of LLM routing as a recommendation system problem is highly innovative and practical. It elegantly transforms the task into a well-understood domain, allowing the use of powerful and mature architectures like Wide & Deep learning. This framing directly addresses the need to balance memorization (specific model strengths) and generalization (handling unseen instructions).
2. The paper tackles a critical real-world challenge in the operationalization of LLMs: cost management. As LLM APIs become ubiquitous, the financial burden is a major concern for developers and businesses. EcoRouter offers a principled and data-driven solution to this problem.
3. A major advantage of EcoRouter is its design for real-world API deployment. By avoiding pre-inference on all candidate models, it minimizes latency. Furthermore, its ability to incorporate new LLMs by generating representations and performing lightweight indexing, rather than full retraining, makes it adaptable to the rapidly evolving LLM landscape, a clear advantage over static Mixture-of-Experts (MoE) models.

Weaknesses

1. The success of EcoRouter is heavily dependent on the massive, pre-computed model-data interaction dataset. Creating and maintaining this dataset requires immense computational resources (running millions of inferences across dozens of models). This presents a significant upfront engineering and financial cost, which may limit the practical adoption of this approach for smaller organizations or research labs.
2. While the paper emphasizes that it avoids the cost of inferring with all candidate LLMs, it is not fully transparent about the computational cost of the router itself. The data representation step involves extracting embeddings and "meta-dimensions" (which may require a separate LLM call), and the recommendation model itself has an inference cost. Quantifying this overhead (e.g., in milliseconds per query) is crucial for a complete cost-benefit analysis.
3. The paper defines cost as per-token cost × total tokens. While a reasonable proxy, real-world LLM API pricing is often more complex, with different rates for input and output tokens, potential costs for specific tool usage, and tiered pricing models. The system's optimality is tied to this simplified cost function.

---

> ### Author Rebuttal · Authors · 2025-07-31
>
> We sincerely thank Reviewer MPGf for the valuable feedback. We have carefully considered each point and provide our responses below:
>
> ### 1. Regarding the high cost of constructing the model-data interaction dataset:
>
> First, we commit to open-sourcing the model-data interaction dataset. The cost of its construction is a one-time, offline expense. In application, this data resource can be continuously and seamlessly collected from interactions between the models and users.
>
> Once the dataset is built and EcoRouter is trained on it, the online routing inference process is extremely efficient. (We have added an experiment **in our response to Reviewer 3Co2** to substantiate this claim).
>
> Furthermore, this construction cost can be amortized. The process is not GPU-intensive (as it primarily involves inference), and most of the data was generated using NVIDIA 4090 GPUs.
>
> * **Handling New Data:** Our routing system can generalize directly to new data as it emerges.
> * **Handling New Models:** When a new model is introduced, our method rapidly adapts by performing inference on a lightweight "bridge" coreset to update the capability distribution. This avoids the need for a full re-inference on all historical data and complete retraining, making the cost of small-scale updates manageable.
>
> To enable the router to accurately learn the "preferences" and capability boundaries of different LLMs, a large-scale, high-quality history of interactions is indispensable. This is crucial for achieving precise and economical routing. We will elaborate on these upfront costs in the final version of the paper.
>
> ### 2. Regarding the computational overhead of the router model itself:
>
> The online inference overhead of EcoRouter primarily stems from two components:
>
> 1.  **Feature Extraction:** We use a small model like `GTE-large` (~0.33B) [307] to extract text embeddings. Its parameter count and computational cost are significantly smaller than any LLM in our candidate pool.
> 2.  **Router Model Inference:** The recommendation framework we employ [327] is essentially a shallow network, making its inference extremely fast with negligible overhead.
>
> We have supplemented our paper with a table detailing the routing latency:
>
> | Metric | Routing Time at Deployment | Qwen2-7B-Instruct Inference Time |
> | :--- | :--- | :--- |
> | Average Latency on GSM8K | ~0.3s | 5-7s |
>
> We will add a dedicated section in the final version to analyze the specific latency and other related overheads of the routing model in detail. Thank you for this valuable suggestion.
>
> ### 3. Regarding the simplification of the cost function:
>
> The formula `Cost = Price_per_token × Total_tokens` is indeed a simplification of real-world API pricing models. We adopted this approach for the following reasons:
>
> * This formula effectively captures the two most critical factors influencing cost: the model's scale (which determines the unit price) and the length of the generated content. Similar proxies are widely used in related work such as FrugalGPT, AutoMix, and Hybrid-LLM.
> * A core strength of EcoRouter is its flexibility concerning the cost function. The function is not hard-coded into the model architecture; it is primarily used to rank the training data (see Equation 2 in the paper). To adapt to more complex pricing strategies (e.g., different pricing for input/output tokens, tiered pricing), we only need to replace the cost function, re-rank the training data accordingly, and retrain the recommendation system. The fundamental methodology and framework remain unchanged.
> * We have added an experiment where cost requirements are provided as a descriptive input. The results are shown in the table below:
>
> | Model Cost Input | Prioritize High Accuracy | Balanced | Prioritize Low Cost |
> | :--- | :--- | :--- | :--- |
> | **EcoRouter (Accuracy, Cost)** | (86.1, 404) | (79.2, 304) | (78.5, 316) |
>
> We will discuss this simplification more explicitly in the limitations section of our paper and emphasize that our framework can be easily adapted to more sophisticated cost models.
>
> ### 4. Regarding the comparison with SOTA LLMs:
>
> * EcoRouter is **not a new LLM** but rather an **LLM routing framework or scheduling system**. Its performance ceiling is determined by the models within its candidate pool.
> * Our framework is designed to **leverage and orchestrate** these SOTA models, not to compete with them. To demonstrate this, we have conducted a new experiment by adding Qwen2.5-14B to our candidate model pool. This integration is lightweight: we only need to perform inference with the new model on the coreset to obtain its capability distribution, without any need for fine-tuning the router. The results are as follows:
>
> | Model | MMLU | TruthfulQA | ARC-C | MMLU-STEM | GSM8K | BBH |
> | :--- | :--- | :--- | :--- | :--- | :--- | :--- |
> | Qwen2-7B-Instruct | 69.1 | 55.5 | 61.4 | 63.5 | 78.9 | 62.9 |
> | Qwen2.5-14B-Instruct | 79.7 | 58.4 | 67.3 | 76.4 | 90.2 | 78.2 |
> | Qwen2-72B-Instruct | 83.8 | 54.9 | 68.6 | 79.9 | 88.9 | 82.9 |
> | **Wide & Deep (Before)** | **82.3** | **67.4** | **71.7** | **77.4** | **86.1** | **78.3** |
> | **Wide & Deep (After)** | **82.0** | **67.7** | **71.7** | **77.5** | **86.8** | **78.1** |
>
> As shown, EcoRouter effectively routes relevant requests (e.g., for GSM8K) to the newly added, more capable Qwen2.5-14B to boost performance. Concurrently, for simpler tasks, it continues to select more economical models, thus maintaining cost-efficiency.
>
> ### 5. Regarding the impact of few-shot sample quantity on performance:
>
> * **Inference Modes as Routing Options:** In EcoRouter, we treat the tuple `(model, inference_mode)` as a holistic candidate for routing. The input features for the router can be augmented with additional information, such as the number of few-shot samples or the reasoning strategy to be used (e.g., CoT, ToT [343]).
> * **Learning the Trade-off:** Increasing the number of few-shot samples generally improves accuracy but also increases the context length, leading to higher inference costs. EcoRouter is specifically designed and trained to learn this **trade-off**.
>
> ***
>
> Once again, we thank you for your valuable time and constructive feedback. We are confident that by incorporating these revisions based on your suggestions, the quality of our paper will be significantly enhanced.

---

> ### Comment · Reviewer_MPGf · 2025-08-01
>
> I am very grateful to the author for his efforts in the rebuttal process. The author's rebuttal partly resolved my doubts (I don't see any explanation about Q1 and Q3), and I decided to maintain the previous rating unchanged.

---

> > ### Comment · Area_Chair_wpmM · 2025-08-05
> >
> > Dear Reviewer MPGf,
> >
> > Thank you for your thoughtful engagement with Paper ID 24785. The authors have expressed a willingness to continue the discussion and address any remaining concerns you may have.
> >
> > I see that authors provided very detailed explanations and results for the questions you asked, but we cannot find your response on the author's rebuttal.
> >
> > If possible, could you kindly elaborate on the reservations that led you to maintain your original rating, especially since your last comment indicated that some doubts were partially resolved? Any additional specific feedback would help facilitate a more productive exchange during the remaining rebuttal period.
> >
> > Given that the deadline for rebuttal period extended, we hope both authors and reviewers continue to engage deeply in the the discussion.
> >
> > Best regards,
> > Area Chair, NeurIPS 2025

---

### Official Review · Reviewer_k8wC · 2025-07-03

**Clarity:** 2
**Significance:** 3
**Originality:** 3
**Rating:** 4
**Confidence:** 4

**Summary:**

This paper works on the model routing task, which is to select from a pool of LLMs the one that can achieve certain tasks while having the lowest cost. The method proposes to represent each LLM with a set of features, including their intrinsic features and performance on evaluation benchmarks. Borrowing ideas from recommender system, a routing model is trained to recommend LLMs for data.

**Questions:**

- It would be better to compare with other model routing methods and conduct more comprehensive analysis on the model performance and cost.
- There are several missing citations in the paper. It would beneficial from some proofreading.
- The definition of Pareto frontier is interesting but it doesn't seem to be relevant to latter discussion.

**Ethical Concerns:**

["NO or VERY MINOR ethics concerns only"]

**Final Justification:**

My concerns are addressed during rebuttal and I have increased the rating to reflect my final judgement

**Limitations:**

yes

**Quality:**

3

**Strengths And Weaknesses:**

### Strengths
- A comprehensive list of LLM features are used to train a recommender model, which looks to be a promising direction for LLM routing problem.
- Experiments show the strong performance of the proposed method.

### Weaknesses
- The experiments could be improved to be more comprehensive. First, there should be inclusion of other routing methods, instead of purely comparing with basic LLMs.
- Also, only reporting the overall performance is not sufficient. Including additional analysis on the model is necessary and often encouraged in the main content.

---

> ### Author Rebuttal · Authors · 2025-07-31
>
> We sincerely thank Reviewer k8wC for their meticulous review of our work. We have added relevant experiments and will present an improved study in the final version of the paper.
>
> ### 1. Regarding the comparison with other model routing methods and a more comprehensive performance and cost analysis:
>
> We have incorporated nine additional comparison methods and seven new benchmarks, including tasks in mathematics, coding, and multi-turn dialogue. This expansion covers all the evaluation metrics considered in RouterBench. For a detailed breakdown, please refer to our response to Reviewer 3Co2.
>
> ### 2. Regarding the missing citations:
>
> Thank you for your careful examination. We have conducted a rigorous proofreading of the entire manuscript and will ensure that all citations are complete and accurate in the final submission.
>
> ### 3. Regarding the definition of the Pareto Frontier and its relevance to the subsequent discussion:
>
> We introduced the concept of the Pareto Frontier to provide a **clear framework and ultimate goal** for the complex optimization problem of LLM routing. An ideal routing system should identify a **Pareto-optimal** solution within the two-dimensional space of "cost-accuracy."
>
> In our experimental tables, we establish a theoretical upper bound by creating a "Best-Performing" baseline—a composite model selecting the optimal model for each specific task. This represents a point on the theoretical Pareto Frontier that prioritizes maximum accuracy.
>
> For instance, in Table 2, the average performance of our method (e.g., Ours w/ DIN at 78.20%) closely approaches the Pareto Frontier (79.39%). Crucially, the **average response length** required to achieve this performance (473) is significantly lower than that of large models like Meta-Llama-3 70B (608).
>
> The table below presents further results demonstrating that by incorporating cost-accuracy control inputs into our routing process, the resulting router can form a superior Pareto Frontier.
>
> | Model Input Preference: | High Accuracy | Balanced | Low Cost |
> | :--- | :--- | :--- | :--- |
> | **Theoretical Pareto Frontier (Accuracy, Cost)** | (88.9, 535) | - | (56.2, 273) |
> | **EcoRouter (Accuracy, Cost)** | (86.1, 404) | (79.2, 304) | (78.5, 316) |
>
> In the final version of the paper, we will include a "cost-accuracy" scatter plot to visually demonstrate how EcoRouter effectively pushes the overall system performance towards the theoretical Pareto Frontier.

---

> > ### Comment · Reviewer_k8wC · 2025-08-05
> >
> > Thanks for the response. The added comparison should be included in the next version to make the experiment more comprehensive. I would like to increase my overall rating.

---

> > > ### Author Response · Authors · 2025-08-05
> > >
> > > Thank you for your valuable feedback. We sincerely appreciate your suggestion.
> > >
> > > We will be sure to incorporate more comprehensive experiments in the final version to ensure our work is more thorough. Thank you again!

---

### Official Review · Reviewer_N3dk · 2025-07-03

**Clarity:** 3
**Significance:** 3
**Originality:** 3
**Rating:** 4
**Confidence:** 4

**Summary:**

The paper introduces EcoRouter, a system that frames LLM routing as a cost-aware recommendation problem. The approach represents candidate models and prompts using a combination of their complexity level, benchmark capabilities, and embedded features. These representations are then used to train a Wide-&-Deep recommender model on a 1B cost-labeled model-instruction pairs. In practice, the system is designed to select the most cost-effective model that satisfies a user-defined accuracy budget. It achieved over 38% in cost savings across six public benchmarks.

**Questions:**

1. The 1B training set is prohibitively expensive for most researchers. What happens if the corpus is 1–2 orders of magnitude smaller?
2. How does EcoRouter perform on RouterBench, especially on coding tasks?

**Ethical Concerns:**

["NO or VERY MINOR ethics concerns only"]

**Limitations:**

This work primarily focus on single-turn prompts, while multi-turn or tool-calling workflows not addressed.

**Paper Formatting Concerns:**

On page8, there is a large blank space between the table and text.

**Quality:**

3

**Strengths And Weaknesses:**

**Strengths**

- By casting the LLM routing problem as a cost-aware recommendation task, the authors align it with well-understood paradigms and avoid the inefficiencies of expensive cascading approaches.
- A key contribution is the introduction of a Pareto-front metric for evaluating cost-accuracy trade-offs which is missing in prior work.
- EcoRouter embeds cost during training by ranking 1 B interaction pairs via a Pareto-front rule. Thus the model is explicitly cost-aware.
- EcoRouter is capable of handling a large set of models and integrating cost directly into the training objective, rather than applying it as a post-hoc filter.
- It also provides a clear strategy for handling new models and out of domain prompts, a "cold-start" problem not fully detailed in relevant works.

**Weaknesses**
- The most significant gap lies in its baseline comparisons. Random choice or embedding-similarity matching are far from the state-of-the-art routing methods. Comparing against competitive works, such as RouteLLM (ref 38 in the paper), Hybrid-LLM (ref 18 in the paper), could strengthen the effectiveness of the method further.
- While the current evaluation covers six public benchmarks, it would be nicer to test on more practical and complexity tasks, like coding, retrieve, and conversational tasks. As a comprehensive benchmark for assessing efficacy of LLM routing system, RouterBench [1] could be a good option to test with.
- The lack of an ablation study on feature groups makes it unclear whether the complex capability vectors offer a real advantage over simpler representations.

[1] Hu, Qitian Jason, Jacob Bieker, Xiuyu Li, Nan Jiang, Benjamin Keigwin, Gaurav Ranganath, Kurt Keutzer and Shriyash Kaustubh Upadhyay. “RouterBench: A Benchmark for Multi-LLM Routing System.” ArXiv abs/2403.12031 (2024): n. pag.

---

> ### Author Rebuttal · Authors · 2025-07-31
>
> We sincerely thank Reviewer N3dk for their recognition of our work's contributions, including: innovatively framing the LLM routing problem as a cost-aware recommendation task, introducing the Pareto-front metric to evaluate the cost-benefit trade-off, and explicitly embedding cost into the training process.
>
> ### 1 & 2. Regarding Baseline Models and Evaluation Benchmarks:
>
> We have added nine new comparative methods and seven new benchmarks (including tasks for mathematics, code, and multi-turn dialogue), taking into account all the evaluations mentioned in RouterBench. For details, please refer to our response to Reviewer 3Co2. We will incorporate the relevant experimental results and analysis in the final version of the paper.
>
> ### 3. Regarding the Ablation Study on Feature Groups:
>
> We conducted a study on combinations of different model features, as shown in the table below:
>
> | Intrinsic Properties | Capability Distributions (Offline Benchmark) | Capability Distributions (Online Evaluation) | EcoRouter Accuracy | Overhead |
> | :--- | :--- | :--- | :--- | :--- |
> | ✓ | | | 79.8 | 496 |
> | | ✓ | | 81.3 | 513 |
> | | | ✓ | 84.0 | 468 |
> | | ✓ | ✓ | 84.2 | 435 |
> | ✓ | ✓ | ✓ | 86.1 | 404 |
>
> *(Note: The checkmark '✓' indicates the feature group was included, and a blank space indicates it was excluded, corresponding to '1' and '0' in the original table respectively.)*
>
> ### 4. Regarding the Scale of Training Data:
>
> We are committed to open-sourcing all of our training data. This data was generated through LLM inference, with the majority of interactions with smaller LLMs being conducted on a 4090 GPU.
>
> Furthermore, training the recommendation model is not as time-consuming or expensive as training a large language model. The entire training process for our reported results took approximately 30 hours on a system with 8x NVIDIA A100 GPUs, indicating that the volume of training data is manageable.
>
> We have supplemented our work with an analysis of routing accuracy under reduced training data sizes:
>
> | Number of Training Samples | 100k | 500k | 1100k (Reported in the paper) |
> | :--- | :--- | :--- | :--- |
> | **Average Performance** | 82.7 | 84.7 | 86.1 |
> | **Average Overhead** | 485 | 468 | 404 |
>
> We will include this sensitivity analysis on the data scale in the final version of the paper. Thank you very much for this valuable suggestion.

---

> > ### Comment · Reviewer_N3dk · 2025-08-04
> > **Thanks**
> >
> > I thank the authors for their detailed response. Your response has fully addressed all of my initial concerns. I am inclined to keep my rating.

---

### Official Review · Reviewer_3Co2 · 2025-07-06

**Clarity:** 3
**Significance:** 4
**Originality:** 3
**Rating:** 4
**Confidence:** 3

**Summary:**

The paper introduces EcoRouter, a recommendation-style router that chooses, at inference time, the “cheapest” large-language model (LLM) and inference mode (direct, self-consistency, CoT, ToT) that will satisfy an accuracy target. Key ingredients are (i) multidimensional representations for both models and prompts (intrinsic metadata + capability “coreset” scores + cross-features), (ii) a Wide-&-Deep/DeepFM-family recommender that learns from >1 B synthetic interactions covering 52 LLMs and 30+ data sets , and (iii) a cost-interval Pareto-frontier metric for evaluation. On six benchmarks EcoRouter reportedly cuts average unit cost by > 38 % with no loss of accuracy and shrinks average output length by ~30 % compared to strong baselines

**Questions:**

please refer to the weaknesses

**Ethical Concerns:**

["NO or VERY MINOR ethics concerns only"]

**Final Justification:**

Thanks for the detailed responses from the authors. I think most of my concerns are well addressed. Considering my score is already positive, I keep my score. Hope the authors could include the new content during rebuttal into the final version.

**Limitations:**

please refer to the weaknesses

**Quality:**

3

**Strengths And Weaknesses:**

Strength:

(1) Timely, practically important problem – production users do face ballooning LLM-API bills; a routing layer that reasons about both model and reasoning depth is valuable.

(2) Clear high-level formulation – turns routing into a constrained optimisation (Eq 1) with a Pareto-front interpretation; keeps accuracy as first-class objective.

(3) Large-scale empirical effort – 50 + LLMs, 3 reasoning modes, 6 diverse data sets; zero-shot tests on unseen models/tasks.

(4) Strong headline numbers – 38 % cost saving at equal accuracy; top accuracy among routing baselines on both general-knowledge (Table 1) and math/BBH (Table 2).

Weakness:

(1) Missing SOTA routers in the comparison. The experiments pit EcoRouter only against Random, GTELarge embedding match, and a few recommender baselines (§4.2 Tables 1–2). Recent cost-aware routers—FrugalGPT, GraphRouter, Hybrid-LLM, RouteLLM—are cited in Related Work but never evaluated, so it is unclear whether EcoRouter’s 38 % cost saving is actually new.

(2) Heuristic cost model. “Token-level costs are estimated from model scale, architecture, etc.” (§3.3, Eq. 2) rather than using real API prices or wall-clock latency. Pricing tiers (e.g., input vs output tokens, 128 k context surcharges) and batching effects are ignored, so the 38 % figure may be optimistic.

(3) No routing-time latency budget. EcoRouter adds three text-encoder passes plus a recommender forward for every query (§3.1/§3.2). The paper does not report milliseconds/query or amortised GPU cost. A router that is cheaper in tokens but slower than simply calling a large model may be unusable.

(4) Over-reliance on scale as a cost proxy. Parameter count (“#Params”) is treated as proportional to cost in Tables 1–2, yet real inference cost is dominated by sequence length and hardware utilisation. For instance, a sparsely-gated MoE (Mixtral-8×22B) costs far less per token than dense 70 B models.

(5) Potential train-test leakage via the 1 415-example ‘bridge’ coreset. The same MMLU categories are used to construct model capability features (§3.1) and to evaluate routing accuracy (Tables 1–2 include MMLU, ARC-C, etc.). If overlap is not carefully filtered, EcoRouter may implicitly see the test distribution during representation learning.

---

> ### Author Rebuttal · Authors · 2025-07-31
>
> We sincerely thank Reviewer 3Co2 for their detailed and insightful review of our work.
>
> We are pleased that the reviewer recognized several core strengths of our work, including the importance of the problem, the clarity of our formulation, and the effectiveness of our experiments.
>
> In the following, we provide detailed explanations and clarifications for the points raised. We are committed to incorporating the corresponding revisions into the final version of our paper:
>
> ### 1. Comparison with SOTA Routers
>
> We have added several SOTA comparison methods, including FrugalGPT, GraphRouter, Hybrid-LLM, and RouteLLM. Furthermore, we have included additional experiments on RouToo, RouterDC, and EmbedLLM.
>
> In addition to the existing six benchmarks, we have expanded our evaluation to include Hellaswag, Winogrande, MMLU-Pro, MATH, MBPP, HumanEval, and MT-Bench. This expansion covers a broader range of tasks, including general knowledge, mathematics, code generation, and multi-turn dialogue.
>
> Our method creates a detailed model representation for the routing process. By using a paradigm of `(model representation + instruction) -> predicted performance`, we consider various aspects of the models, such as their intrinsic information and capability distributions. In a sense, this approach encapsulates the principles of many of the compared methods.
>
> Our replication details are as follows:
>
> 1.  **FrugalGPT (Cascade Strategy)**
>     * **Routing Mechanism**: This method employs a cascade calling and post-generation scoring mechanism. It does not predict performance but rather evaluates the generated answer after the fact.
>         * **Input**: `(generation result from the current model, original instruction)`. The output is similar to a reward model, assessing whether the result is satisfactory.
>     * **Replication Details**: We constructed an LLM cascade using `Qwen2-7B-Instruct`, `Qwen2-57B-A14B-Instruct`, and `Qwen2-72B-Instruct`. For the scoring function, we used the DistilBERT model from their official code, retrained on our own data. To ensure evaluation stability, we did not use the "prompt adaptation" or "LLM approximation" (caching) strategies from the original paper.
>
> 2.  **GraphRouter**
>     * **Routing Mechanism**: Treats routing as an edge prediction task on a graph.
>         * **Input**: The instruction and a description of the models.
>     * **Replication Details**: Due to limitations in the official implementation regarding multi-GPU distribution for single-graph data, we completed the replication using a subset of the training data.
>
> 3.  **Hybrid-LLM**
>     * **Routing Mechanism**: DeBERTa + a classifier.
>         * **Input**: The instruction. The model representation consists of the parameters of each DeBERTa classifier.
>
> 4.  **RouteLLM**
>     * We primarily replicated its two core strategies (the other two are similar to Hybrid-LLM):
>         * **Similarity-Weighted Ranking**: **Input** is a `new instruction`. It is compared against historical data and modeled using the Bradley-Terry model. **Output** is a `full ranking of the models`.
>         * **Matrix Factorization**: **Input** is `(instruction embedding, model embedding)`, where the model embedding is derived from the factorization of a historical performance matrix. **Output** is a `predicted performance score`.
>
> 5.  **Methods from RouterBench**
>     * We replicated its non-cascade routers:
>         * **k-Nearest Neighbors (KNN) Router**: **Input** is the `embedding vector of the request`. **Output** is a `performance estimate based on the average performance of the K-nearest neighbors` (where historical performance data serves as the model representation).
>
> 6.  **Routoo**
>     * **Routing Mechanism**: Uses a decoder-only LLM to process the input, combined with a unique model embedding vector.
>         * **Input**: `(instruction, target model ID)`. The model ID maps to a learnable "model embedding." We did not adopt this user-item style of learnable representation because it would require incremental retraining of the entire routing structure whenever a new model is added.
>
> 7.  **RouterDC**
>     * **Routing Mechanism**: Trains the router based on contrastive learning.
>         * **Input**: `(instruction, learnable model embedding)`.
>
> 8.  **EmbedLLM**
>     * **Routing Mechanism**: Learns compact representations of models via matrix factorization.
>         * **Input**: The instruction and relevant historical performance data.
>
> | Model | MMLU | TruthfulQA | ARC-C | MMLU-STEM | GSM8K | BBH |
> | :--- | :--- | :--- | :--- | :--- | :--- | :--- |
> | Qwen2-7B-Instruct | 69.1 | 55.5 | 61.4 | 63.5 | 78.9 | 62.9 |
> | Qwen2-72B-Instruct | 83.8 | 54.9 | 68.6 | 79.9 | 88.9 | 82.9 |
> | Random Selection | 73.0 | 58.9 | 67.8 | 68.0 | 77.7 | 73.6 |
> | Wide & Deep | 82.3 | 67.4 | 71.7 | 77.4 | 86.1 | 78.3 |
> | DIN | 83.9 | 66.8 | 73.0 | 78.4 | 87.2 | 79.9 |
> | FrugalGPT | 81.8 | 63.6 | 72.6 | 73.1 | 83.2 | 77.5 |
> | RouterBench | 72.3 | 62.0 | 71.9 | 70.2 | 80.1 | 74.7 |
> | Hybrid-LLM | 74.1 | 59.6 | 69.3 | 70.8 | 77.0 | 73.3 |
> | RouteLLM (Bradley-Terry) | 79.0 | 62.8 | 69.6 | 71.0 | 81.7 | 74.1 |
> | RouteLLM (Matrix Dec.) | 77.5 | 60.8 | 66.5 | 70.7 | 81.3 | 74.5 |
> | GraphRouter | 80.2 | 59.8 | 69.7 | 75.3 | 80.4 | 75.8 |
> | ROUTOO | 80.9 | 63.6 | 71.7 | 75.5 | 83.2 | 75.2 |
> | RouterDC | 81.7 | 64.2 | 73.2 | 75.0 | 81.8 | 77.2 |
> | EmbedLLM | 82.0 | 62.4 | 72.3 | 70.6 | 81.6 | 76.4 |
>
> ### 2. Cost in Relation to Real API Pricing
>
> Our cost model is based on estimations of model scale, architecture, and generation length. Our initial goal was to establish a **general and relatively stable** framework for measuring cost that does not directly depend on the frequently changing and complex pricing rules of specific API providers (like OpenAI or Google). This approach ensures our conclusions have broader applicability.
>
> Due to network instability, we have not yet completed supplementary experiments with real APIs. However, in the final version, we will purchase API access for the relevant models (with their real-world pricing) and include a comparison of the actual costs incurred.
>
> ### 3. Lack of Discussion on Routing Latency
>
> The online inference overhead of EcoRouter primarily comes from two components: 1) **Feature Extraction**: We use a small model like `GTE-large` (~0.33B) to extract text embeddings. Its parameter count and computational cost are significantly smaller than any LLM in the candidate pool. 2) **Recommendation Model Inference**: The recommendation framework we employ is essentially a shallow network, making its inference extremely fast with negligible overhead.
>
> We have added the routing latency in the table below:
>
> | Average Latency on GSM8K | EcoRouter Routing Time | Qwen2-7B-Instruct Inference Time |
> | :--- | :--- | :--- |
> | Time | ~0.3s | 5~7s |
>
> ### 4. Over-reliance on Scale as a Cost Proxy
>
> We would like to clarify our cost calculation. In our framework, the unit token cost is estimated based on the model's scale. The total cost is then calculated as the product of this **"model's unit token cost"** and the **"length of the generated output (in tokens)"**.
>
> We will update the paper to make this clearer in the final version, particularly in the tables. We appreciate this valuable suggestion.
>
> ### 5. Potential Train-Test Data Leakage
>
> Our "bridge coreset" and the evaluation test sets are completely disjoint. We will state this more explicitly in **§3.1** and **§4.1** of the paper.
>
> Furthermore, we've observed that the learning difficulty increases significantly when there is a large distributional shift (measured by the Wasserstein distance) between the historical performance dataset and the target evaluation set.
>
> | Distributional Distance (Wasserstein) between Historical Data and GSM8K | EcoRouter Learning Difficulty (Convergence Steps) |
> | :--- | :--- |
> | 6.29 | ~2k |
> | 38.41 | ~5k |
> | 53.2 | ~8k |
>
> Once again, we thank you for your valuable feedback. We will continue to work diligently to improve our paper.

---

### Author Response · Authors · 2025-08-09

Dear ACs and Reviewers,

As the discussion period draws to a close, we would like to sincerely thank you again for your time and insightful feedback. We hope our responses and clarifications have been helpful in addressing your valuable comments.

We are committed to improving our work based on your suggestions. In the final version, we will incorporate all the details and utilize the additional page to thoroughly address them.

We greatly appreciate your consideration and hope our work earns your support.

Best regards,\
The Authors

---

### Decision · Program_Chairs · 2025-09-17

**Decision:**

Accept (poster)

**Comment:**

This paper introduces EcoRouter, a recommendation-system-based framework for routing user instructions to the most cost-effective LLM in a candidate pool. By constructing multidimensional model and data representations (intrinsic properties, benchmark capability scores, and cross features) and training a Wide&Deep/DeepFM-style recommender on >1B synthetic interactions, EcoRouter learns to predict which (model, inference mode) pairing best satisfies a user-defined accuracy target under cost constraints. A Pareto frontier–based evaluation metric is also proposed. Empirically, EcoRouter reduces costs by >38% on six benchmarks across 50+ LLMs while maintaining accuracy.

Strengths. Reviewers agreed the framing of routing as a recommendation problem is both novel and practically important (Reviewer k8wC, N3dk, MPGf). The introduction of a Pareto-frontier cost–accuracy metric was noted as a methodological contribution (Reviewer N3dk). The large-scale evaluation across diverse LLMs and reasoning modes demonstrated strong headline improvements (Reviewer 3Co2). Practical deployment advantages, such as avoiding inference on all candidate models and adaptability to new LLMs, were emphasized as real-world strengths (Reviewer MPGf).

Weaknesses. Several reviewers highlighted limitations in evaluation: missing competitive baselines such as RouteLLM, GraphRouter, Hybrid-LLM, and FrugalGPT (Reviewer 3Co2, N3dk, k8wC); absence of RouterBench coding/multi-turn benchmarks (Reviewer N3dk); and limited ablation analyses (Reviewer N3dk, k8wC). Cost modeling was flagged as simplified and heuristic, not tied to real API pricing or latency (Reviewer 3Co2, MPGf). Concerns about router overhead and whether cost savings persist in real deployments were raised (Reviewer 3Co2, MPGf). Reviewer 3Co2 also questioned possible train–test leakage from the MMLU-based coreset. Finally, the approach requires a massive precomputed interaction dataset (>1B samples), raising barriers for replication (Reviewer MPGf).

Discussion and Rebuttal. The authors’ rebuttal was substantive and constructive. They added nine missing baselines (FrugalGPT, GraphRouter, Hybrid-LLM, RouteLLM, etc.) and seven new benchmarks (including coding and dialogue), as well as ablations on feature groups and sensitivity to training data size. They clarified that router overhead is ~0.3s vs. 5–7s for a 7B LLM, and committed to adding real API pricing experiments. They also confirmed no leakage between the bridge coreset and test sets. Reviewers largely accepted these clarifications: Reviewer k8wC explicitly raised their score; Reviewer N3dk stated all concerns were fully addressed; Reviewer 3Co2 maintained a borderline-accept rating but acknowledged improvements; Reviewer MPGf appreciated the responses but held their borderline-reject score, citing only partial resolution of concerns.

Decision. While one reviewer remains unconvinced, the majority of reviewers shifted positively after rebuttal, recognizing the paper’s strong empirical contribution, novel framing, and practical significance for LLM deployment. The weaknesses (e.g., dataset construction cost, proxy cost model) are real but do not undermine the central technical contributions. Given its relevance to scalable and economical LLM usage—a timely and practically pressing problem—I recommend acceptance. However, due to residual concerns about evaluation completeness and cost realism, I do not recommend spotlight/oral; the work is valuable but not yet fully mature in evaluation rigor.

Follow-up for Camera-Ready. For the final version, please:
1. Incorporate the newly added baselines (FrugalGPT, GraphRouter, Hybrid-LLM, RouteLLM, etc.) and expanded benchmarks (e.g., coding, dialogue) into the main paper.
2. Provide empirical results under real API pricing and latency measurements, not only heuristic proxies, to strengthen cost-effectiveness claims.
3. Explicitly report routing overhead in ms/query and clarify the amortized cost relative to direct large-model inference.
4. Clearly document how the bridge coreset is separated from evaluation datasets to avoid any impression of train–test leakage.
5. Discuss the scalability and accessibility of the required 1B+ interaction dataset, and outline strategies for researchers with limited resources (e.g., smaller-scale training feasibility).
6. Add a short discussion of broader limitations, such as reliance on single-turn prompts and the simplified cost definition, as flagged by reviewers.

Recommendation: Accept.